

# Comparing impacts of metal contamination on macroinvertebrate and fish assemblages in a northern Japanese river

Hiroki Namba[1,2,*], Yuichi Iwasaki[3,*], Kentaro Morita[4,5], Tagiru Ogino[6], Hiroyuki Mano[3], Naohide Shinohara[3], Tetsuo Yasutaka[7], Hiroyuki Matsuda[1] and Masashi Kamo[3]

[1] Graduate School of Environment and Information Sciences, Yokohama National University, Yokohama, Kanagawa, Japan
[2] Nippon Koei, Tokyo, Japan
[3] Research Institute of Science for Safety and Sustainability, National Institute of Advanced Industrial Science and Technology, Tsukuba, Ibaraki, Japan
[4] Japan Fisheries Research and Education Agency, Sapporo, Hokkaido, Japan
[5] Field Science Center for Northern Biosphere, Hokkaido University, Horokanai, Hokkaido, Japan
[6] Hokkaido Research Organization, Sapporo, Hokkaido, Japan
[7] Geological Survey of Japan, National Institute of Advanced Industrial Science and Technology, Tsukuba, Ibaraki, Japan
[*] These authors contributed equally to this work.

Corresponding author
Yuichi Iwasaki, yuichiwsk@gmail.com

## ABSTRACT

Researchers have long assessed the ecological impacts of metals in running waters, but few such studies investigated multiple biological groups. Our goals in this study were to assess the ecological impacts of metal contamination on macroinvertebrates and fishes in a northern Japanese river receiving treated mine discharge and to evaluate whether there was any difference between the metrics based on macroinvertebrates and those based on fishes in assessing these impacts. Macroinvertebrate communities and fish populations were little affected at the downstream contaminated sites where concentrations of Cu, Zn, Pb, and Cd were 0.1–1.5 times higher than water-quality criteria established by the U.S. Environmental Protection Agency. We detected a significant reduction in a few macroinvertebrate metrics such as mayfly abundance and the abundance of heptageniid mayflies at the two most upstream contaminated sites with metal concentrations 0.8–3.7 times higher than the water-quality criteria. There were, however, no remarkable effects on the abundance or condition factor of the four dominant fishes, including masu salmon (*Oncorhynchus masou*). These results suggest that the richness and abundance of macroinvertebrates are more sensitive to metal contamination than abundance and condition factor of fishes in the studied river. Because the sensitivity to metal contamination can depend on the biological metrics used, and fish-based metrics in this study were limited, it would be valuable to accumulate empirical evidence for ecological indicators sensitive to metal contamination within and among biological groups to help in choosing which groups to survey for general environmental impact assessments in metal-contaminated rivers.

## INTRODUCTION

The impact of trace metals on aquatic ecosystems is an important issue in many regions of the world (*Nriagu & Pacyna, 1988*; *Iwasaki & Ormerod, 2012*). Laboratory toxicity tests of surrogate species are routinely used to assess the potential effects of metals on aquatic organisms and to provide a first step in inferring the effects on ecosystems. Responses of surrogate species in the laboratory, however, are not necessarily a good indicator for predicting responses of natural populations and communities because of, for example, the short test durations and failures to consider sensitive life stages, dietary exposure, and/or interspecies interactions (*Kimball & Levin, 1985*; *Niederlehner et al., 1990*; *Hickey & Clements, 1998*; *Clements, Cadmus & Brinkman, 2013*; *Iwasaki, Schmidt & Clements, 2018*). Thus, biological assessments of natural aquatic populations and communities that likely reflect time-integrated effects can provide useful information for evaluating ecological impairments in actual environments (*Barbour et al., 1999*).

In conducting the biological assessments in natural environments, the first question to answer is which aquatic organisms are to be investigated. For example, benthic macroinvertebrates have a wide range of sensitivities to contamination by metals (*Iwasaki, Schmidt & Clements, 2018*). Also, macroinvertebrates have been the most frequently used in assessing the ecological impacts of metals in streams and rivers (*Namba et al., 2020*). Studies have indicated, however, that in aquatic ecosystems there are generally low correlations between changes in different biological groups (*Heino, 2010*; *De Morais et al., 2018*; *Namba et al., 2020*). Despite this observation, a surprisingly limited number of studies published in peer-reviewed journals have investigated multiple biological groups in metal-contaminated rivers (*Freund & Petty, 2007*; *Namba et al., 2020*). Therefore, to provide a more comprehensive assessment for overall ecosystem protection, it is important to investigate responses of not only macroinvertebrates but also other biological groups such as fishes in metal-contaminated rivers and to accumulate such case studies to better understand the ecological impacts of metal contamination on different biological groups.

The closed Motokura mine is located in the upstream area of the Tokushibetsu River in northern Japan (Fig. 1). The mine mainly produced Cu, Pb, and Zn. In 1962, there were mass mortalities of Pacific salmon (*Oncorhynchus* spp.) in the river and *Takayasu et al. (1964)* concluded that mine drainage discharged into the river was likely a major cause. Even after the mine closure in 1967, treated mine drainage has been perennially discharged into a small tributary of the contaminated river (Fig. 1), which provides a unique opportunity to investigate any ecological consequences of the long-term exposure. A bioassessment in 2017 using only macroinvertebrates showed that the abundance and richness of macroinvertebrates were little affected at downstream sites in the Tokushibetsu River (*Iwasaki et al., 2020b*). Given that recreational fishing for salmonids is popular in this region (Hokkaido Island, Japan; *Miyakoshi et al., 2009*) and hatchery-reared masu salmon are released into the river system, it is important to evaluate the effects of mine drainage

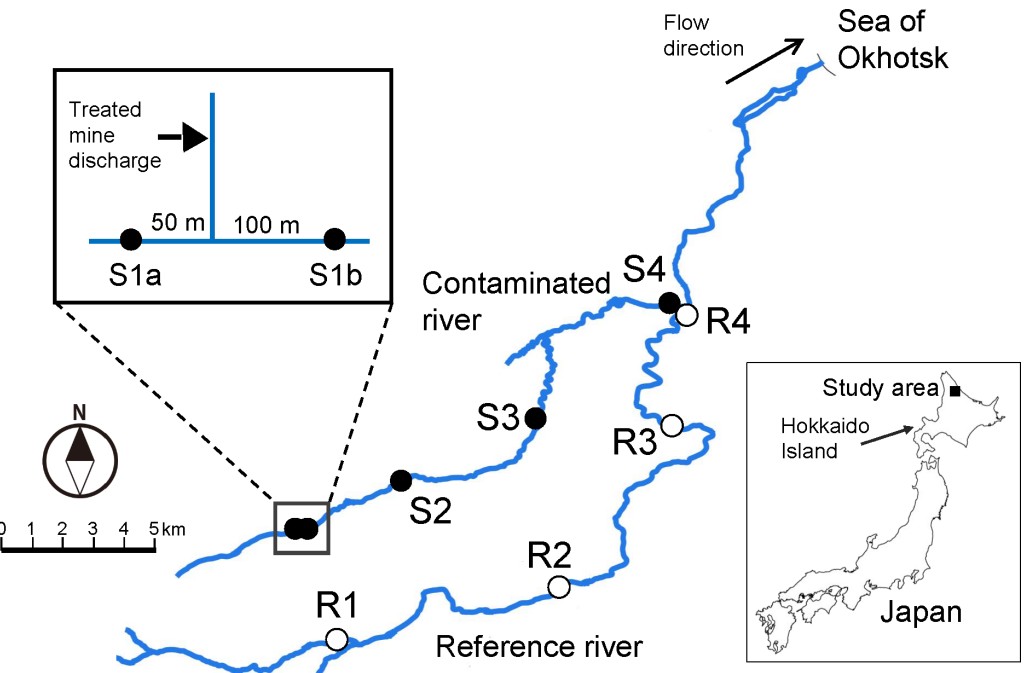

**Figure 1** **Map showing location of the study area and sampling sites.** Map was created using Quantum Geographic Information System (QGIS version 3.10; http://qgis.osgeo.org) based on National Land Numerical Information provided by Geospatial Information Authority of Japan (http://nlftp.mlit.go.jp/ksj/).

on not only macroinvertebrates as food resources for fish, but also on fish communities. However, no recent studies have evaluated the effects of mine discharge on fish in the river (but see *Takayasu et al., 1964*). We thus aimed to assess whether there are ecological impacts in the contaminated river by investigating macroinvertebrates and fishes. By doing so, we also evaluated whether there were any differences between metrics based on macroinvertebrates and those using fishes in detecting the effects of metal contamination.

## MATERIALS & METHODS

### Study site

Field sampling of macroinvertebrates, fishes, and physicochemical characteristics was performed at nine sites in the Tokushibetsu River system on Hokkaido Island, northern Japan (Fig. 1) from 26 to 28 June 2018. Except for the fish survey, the methods adopted were described for a previous study in June 2017 by *Iwasaki et al. (2020b)*. The basin area of the Tokushibetsu River system is approximately 300 km$^2$, and the predominant land uses are forest (88%) and agriculture (<10%; mainly meadow), with little urban use (<1%) according to *Iwasaki et al. (2020a)*. The geology of the studied catchment is characterized mainly by igneous rocks, followed by sedimentary rocks (*Geological Survey of Japan, 2015*).

Five of the nine sites (sites S1a–S4) were in the Ofuntarumanai River, a metal-contaminated stream receiving treated mine discharge, and four reference sites (R1–R4) were in the main stream of the Tokushibetsu River. The reference sites, with similar

**Table 1 Physical parameters at the study sites in the Tokushibetsu River system, northern Japan.**

| Site | Elevation (m a.s.l.) | Catchment area (km²) | Channel width (m) | Studied riffles | | | Sampled stones | | |
|------|------|------|------|------|------|------|------|------|------|
| | | | | Width (m) | Maximum depth (cm) | Maximum velocity (cm/s) | Depth (cm) | Velocity (cm/s) | Relative surface area (cm²) |
| Contaminated sites | | | | | | | | | |
| S1a | 330 | 18 | 11 | 4.5 | 27 | 170 | 6.9 (3.5) | 102 (33) | 1026 (403) |
| S1b | 330 | 19 | 10 | 8.6 | 28 | 170 | 6.3 (3.2) | 73 (27) | 851 (169) |
| S2 | 230 | 29 | 9 | 9 | 25 | 165 | 7.6 (3.6) | 98 (25) | 1003 (260) |
| S3 | 130 | 46 | 11 | 14 | 25 | 200 | 6.6 (2.5) | 98 (31) | 1192 (404) |
| S4 | 30 | 117 | 21 | 5.1 | 25 | 230 | 6.5 (2.8) | 89 (24) | 1114 (396) |
| Reference sites | | | | | | | | | |
| R1 | 285 | 27 | 11 | 11 | 26 | 180 | 7.4 (1.2) | 87 (47) | 1016 (278) |
| R2 | 170 | 77 | 14 | 11 | 25 | 170 | 4.2 (1.6) | 91 (40) | 931 (311) |
| R3 | 75 | 107 | 21 | 16 | 23 | 170 | 6.4 (3.3) | 100 (36) | 1007 (280) |
| R4 | 35 | 127 | 24 | 7.7 | 24 | 170 | 5.5 (1.7) | 86 (24) | 1039 (322) |

**Notes.**
Depth, velocity, and relative surface area for sampled stones are the means (and standard deviations) of five stones sampled.

physicochemical characteristics other than metal contamination (e.g., channel width), were established at similar elevations as the contaminated sites to avoid problems caused by natural longitudinal changes in community structure (*Clements, 1994*; *Tokeshi, 2009*; *Iwasaki et al., 2012*; *Morita, Sahashi & Tsuboi, 2016*). Study sites with the same numbers had similar elevation levels, for example S1 (a and b) and R1 (Table 1). These study sites were on third- or fourth-order rivers. Sites S1a and S1b were upstream and downstream of the inflow of treated mine discharge, respectively (Fig. 1). Because the treated discharge was not the sole source of metal contamination in the Ofuntarumanai River (*Iwasaki et al., 2020b*) and there was a contaminated tributary upstream of S1a (*Takayasu et al., 1964*), we regarded S1a as a contaminated site (see below for more details). Permits for field sampling in the river were obtained from the local municipal office and Hokkaido government.

Similar field sampling was performed at the same nine sites in the Tokushibetsu River system on 26 and 27 September 2018. This field sampling was carried out by the Japanese Ministry of Economy, Trade and Industry, and the results are publicly available (see Supplemental materials for more details). The field sampling method for benthic macroinvertebrates in this survey was different from that in the present study and no replicates were taken at seven of the nine sites; thus, it was difficult to simultaneously analyze the field data collected in both June and September 2018. Instead, we note the results of the additional field study in the Discussion section to evaluate our findings based on the field sampling conducted in June 2018.

## Water-quality parameters

During field sampling, three water samples (50 ml) were filtered from each study site for dissolved metals analysis (0.45 μm pore-size) and refrigerated in the field. Ultrapure nitric

acid was added to those water samples on the day of sampling so that the pH was less than 2. Concentrations of dissolved Cu, Zn, Cd, and Pb were measured by using an inductively coupled plasma mass spectrometer (Element XR, Thermo Fisher Scientific, Tokyo, Japan) according to method 200.8 of the U.S. Environmental Protection Agency (*U.S. EPA, 1994*). The limits of quantification were 0.001 μg/L for Cu, 0.06 μg/L for Zn, and 0.005 μg/L for both Cd and Pb.

Water temperature, dissolved oxygen, pH, and electrical conductivity were measured by using multi-parameter portable meters (Multi 3630IDS, Xylem Analytics Germany, Weilheim, Germany). Filtered water samples were also collected for measuring concentrations of dissolved organic carbon (DOC) and major ions ($Na^+$, $K^+$, $Ca^{2+}$, $Mg^{2+}$, $Cl^-$, and $SO_4^{2-}$). DOC was measured with a total organic carbon analyzer (TOC-L CPH, Shimadzu, Kyoto, Japan). Concentrations of major ions were measured with an ion chromatograph (Dionex ICS-1100/2100, Thermo Fisher Scientific). We calculated water hardness as $2.497 \times [Ca^{2+}] + 4.118 \times [Mg^{2+}]$.

As an index of contamination by multiple metals, we calculated the cumulative criterion unit (CCU; *Clements et al., 2000*) as the sum of the ratios of measured concentrations of four metals to the US EPA hardness-adjusted water-quality criteria for aquatic life (WQC; *U.S. EPA, 2002*) because the water quality standards for aquatic life are available only for Zn in Japan:

$$CCU = \sum (m_i/c_i) \tag{1}$$

where $m_i$ is the concentration of dissolved metal $i$ and $c_i$ is the corresponding WQC. Hardness-adjusted WQC for Cu, Zn, Cd, and Pb were calculated at a water hardness of 10 mg/L based on the observed range of water hardness in this study (Table 2) and a previous study of the same river (*Iwasaki et al., 2020b*). Note that, because the hardness of 10 mg/L is below the lower end of the hardness range of toxicity data used in the WQC development (20 mg/L; *U.S. EPA, 2002*), caution is required for the interpretation of the calculated CCU values. Also, we did not consider water quality variables other than water hardness (e.g., pH and DOC) in this calculation (*Iwasaki et al., 2020b*). This is because these variables varied little among study sites (Table 2), and US EPA WQC based on biotic ligand models that can consider the influence of water chemistry on metal toxicity were available only for Cu (*U.S. EPA , 2007*).

## Physical parameters

Average channel width (surface-water width measured at run) and riffle width were measured at each study site. Riffle width was averaged if benthic macroinvertebrates were collected at multiple riffles within individual sites. The catchment area of each site was quantified using a digital elevation model (50-m grid; Geographical Survey Institute of Japan, http://www.gsi.go.jp/ENGLISH/index.html) and a geographic information system (ArcGIS 10.2 for Desktop, Esri Japan, Tokyo, Japan). Maximum water velocity and depth were evaluated on the basis of measurements at multiple places in riffles from which macroinvertebrates were collected at each study site. Current velocity was measured at 60% of water depth using an electromagnetic velocity meter (VR-301; Kenek, Tokyo, Japan).

**Table 2 Water-quality measurements at study sites in the Tokushibetsu River system, northern Japan (26–28 June 2018).**

| Site | Cu | Cd | Pb | Zn | CCU | Temp | pH | DO | DOC | Conductivity | Hardness |
| --- | --- | --- | --- | --- | --- | --- | --- | --- | --- | --- | --- |
| | | | Dissolved (μg/L) | | | (°C) | | (mg/L) | (mg/L) | (μs/cm) | (mg/L) |
| Contaminated sites | | | | | | | | | | | |
| S1a | 1.0 | 0.13 | 0.69 | 24.0 | 8.4 | 9.1 | 7.1 | 11 | 0.3 | 54 | 13 |
| S1b | 1.1 | 0.16 | 0.71 | 27.5 | 9.4 | 9.3 | 7.0 | 11 | 0.4 | 52 | 13 |
| S2 | 0.8 | 0.17 | 0.25 | 25.9 | 6.8 | 9.4 | 7.2 | 11 | 0.3 | 57 | 14 |
| S3 | 0.5 | 0.07 | 0.23 | 11.5 | 3.8 | 11.5 | 7.4 | 11 | 0.4 | 56 | 13 |
| S4 | 0.3 | <0.005 | 0.05 | 4.8 | 0.9 | 10.2 | 7.5 | 11 | 0.7 | 60 | 14 |
| Reference sites | | | | | | | | | | | |
| R1 | 0.1 | <0.005 | 0.09 | 25.3 | 2.1 | 10.6 | 7.5 | 10 | 0.8 | 41 | 10 |
| R2 | 0.1 | <0.005 | <0.005 | 0.1 | 0.1 | 10.2 | 7.5 | 11 | 0.7 | 46 | 11 |
| R3 | 0.1 | <0.005 | 0.04 | 0.1 | 0.3 | 11.7 | 7.7 | 11 | 0.6 | 48 | 11 |
| R4 | 0.1 | <0.005 | 0.03 | 0.3 | 0.3 | 9.7 | 8.0 | 12 | 0.7 | 50 | 12 |
| WQC | 1.3 | 0.05 | 0.19 | 16.8 | | | | | | | |

**Notes.**

DO, dissolved oxygen; DOC, dissolved organic carbon; CCU, cumulative criterion unit (see text for details); Temp, temperature; WQC, U.S. EPA chronic water-quality criterion at a water hardness of 10 mg/L (*U.S. EPA, 2002*).

Limits of quantification for Cu, Zn, Cd, and Pb were 0.001, 0.06, 0.005, and 0.005 μg/L, respectively.

## Macroinvertebrates

At cobble-dominated riffles at each site, we collected macroinvertebrates from five randomly chosen stones (maximum diameter, 14–27 cm) using a Surber net (mesh size, 0.355 mm). Samples were preserved in the field in 99.5% ethanol and washed through a 0.5-mm sieve in the laboratory. Macroinvertebrates remaining on the sieve were preserved in 70% ethanol and identified generally to genus or species level. For each stone from which macroinvertebrates were collected, water depth and current velocity (at 60% depth) were measured above its upper surface before collecting macroinvertebrates. The relative surface area of each stone was estimated as the product of its maximum diameter and maximum boundary length. Although the mesh size of the sieve used in this study (0.5 mm) is commonly adopted in nationwide samplings of benthic macroinvertebrates in Japan, the smaller mesh size (i.e., 0.35 mm) can collect smaller individuals, which are more sensitive to metal contamination than larger ones (*Cadmus et al., 2020*). However, because our sampling was performed in late spring, when most insect individuals are expected to be at late larval stages and larger than 0.5 mm, the use of the smaller mesh size should not have materially changed our results.

We analyzed eight community metrics for abundance (the number of individuals per stone) and richness (the number of taxa per stone): total abundance, total taxon richness, and the abundance and richness of three major aquatic insect orders in the benthic samples collected: Ephemeroptera (mayflies), Trichoptera (caddisflies), and Diptera (true flies). We also determined the abundance of the dominant families (i.e., Ephemerellidae, Baetidae, Heptageniidae, Hydropsychidae, Chironomidae, and Simuliidae) of the three major aquatic groups, which were defined as those families that accounted for more than 5% of the total abundance at each sampled stone and that were collected at more than 30% of the sampled stones (i.e., more than 14 stones of a total of 45 stones collected). For all macroinvertebrate

metrics, the means and standard errors (as indicators for the uncertainty in site mean) of five stones at each site were calculated and used for further analyses. Macroinvertebrate abundances were $\log_{10}$-transformed $(x + 1)$ before calculation of the site means to satisfy the assumptions of further analyses.

## Fishes

At each site, we established five fish-sampling areas of approximately 5 m × 10 m to cover all of the habitats available (e.g., run, riffle, pool, and backwater) as much as possible. The distance between sampling areas was set to be >20 m. Fishes were collected from the downstream to the upstream end of each sampling area by using a backpack electrofishing unit (200–300 VDC; LR-20B, Smith-Root, Inc., Vancouver, WA, USA) and by throwing a cast-net. After one pass electrofishing, we used a cast-net four or five times within each sampling area to catch fishes in places where the pool was too deep for electrofishing to work. For the fish collection, we did not use block nets because it was impractical to effectively put them in place without disturbance. However, by expending a certain amount of effort, our sampling methods should be acceptable for comparing the abundances of fish communities. The captured fishes were anesthetized with phenoxyethanol and identified to species level if possible. The fork length was measured to the nearest one mm and body weight was measured to the nearest 0.1 g onsite.

A total of five fish species were collected: *Oncorhynchus masou* (masu salmon; Salmonidae), *Salvelinus leucomaenis* (white-spotted char; Salmonidae), *Barbatula oreas* (stone loach; Nemacheilidae), *Lethenteron* spp. (lamprey; Petromyzontidae), and *Tribolodon* spp. (Cyprinidae). We excluded *Tribolodon* spp. from the analyses because of their very limited abundance in our samples (only two individuals collected at R4) and determined the abundance (the number of individuals per sampling area) and condition factor of the other four species. The abundances of fishes were $\log_{10}$-transformed $(x + 1)$, and the means and standard errors of the five replicate samplings at each site were used for later analyses. Also, the condition factor (CF) was calculated as an indicator representing the health status of individual fish by using the following equation:

$$CF = \text{body weight(g)}/[\text{fork length(cm)}]^3 \times 1000. \tag{2}$$

The condition factor is relatively easy to measure in the field and is a sensitive measure for detecting population-level consequences of metal contamination (*Munkittrick & Dixon, 1989a*; *Environment and Climate Change Canada, 2015*). Condition factor data were pooled at individual sites and used in later analyses.

Approximately 128,000 individual hatchery-reared masu salmon fry (*O. masou*; mean fork length: 5.6 cm) were released at a location between S1b and S2 on the contaminated river on 6 June 2018. Masu salmon were also released at three other locations including a tributary between R1 and R2 in the Tokushibetsu River basin in April and June 2018. All released fry had thermally induced otolith marks (*Volk, Schroder & Grimm, 1999*). To estimate the proportion of wild (natural-origin) and hatchery fish at each site, we sampled and checked the otolith marks of 20–27 masu salmon captured from each site in the laboratory. We then tested whether the inclusion of hatchery fish affected the results of our analyses.

## Data analysis

All statistical tests were performed using R version 3.6.1 (*R Core Team, 2019*). A significance level ($\alpha$) of 0.05 was used. All the data used are available in the Supplementary File. In order to evaluate any effects at the five contaminated sites in the river receiving the mine discharge (i.e., S1a–S4), we first evaluated whether the site mean for each biological metric was within the 90% prediction interval calculated by fitting an intercept-only linear regression model to the reference site means. We refer to the 90% prediction intervals as "reference ranges" that are assumed as likely observed ranges of means at reference sites. We then examined whether there were statistically significant differences in biological metrics between each contaminated site and the corresponding reference site with a similar elevation (R1 vs. S1a, R1 vs. S1b, R2 vs. S2, R3 vs. S3, R4 vs. S4) by using a multiple comparison test (the single-step *P*-value adjustment; *Bretz, Hothorn & Westfall, 2010*) after analysis of variance.

We used the results of these two analyses to operationally interpret the findings in three ways. If the mean of a given biological metric at a contaminated site was lower or higher than the corresponding reference range and was significantly lower or higher than that of the corresponding reference site by the multiple comparison test, we report that as an "adverse effect". If either one of these two results was observed, we report that as "some effect of concern" and if neither was observed, we conclude that there was "no effect of concern."

# RESULTS

## Physicochemical parameters

Concentrations of the four trace metals (Cu, Zn, Cd, and Pb) at the contaminated sites (S1a–S4) were approximately 2 to 190 times higher than the concentrations at the corresponding reference sites at similar elevations, except for the concentration of Zn (25 µg/L) at reference site R1, which was similar to the concentrations at S1a and S1b (Table 2). Concentrations of the metals excluding Cu at many contaminated sites were higher than the values of the US EPA hardness-adjusted WQC for aquatic life, with higher concentrations and CCU values at the upstream sites. As previously observed (*Iwasaki et al., 2020b*), there was little difference in metal concentrations between the site just upstream (S1a) and just downstream (S1b) of the inflow of treated discharge. This is most likely due to the high concentrations of metals in an upstream tributary draining the mining area (Iwasaki et al., personal observations, 2019; this is beyond the scope of the present study). CCU values were greater than 1 at all of the contaminated sites except for S4, indicating potential ecological risks based solely on the concentrations of the trace metals measured.

There were marginally lower values of pH, DOC, and water hardness at the metal-contaminated sites compared with reference sites (Table 2), all of which generally increase the bioavailability of metals (*Adams et al., 2020*). The estimated catchment areas of the metal-contaminated sites were generally larger than those of the corresponding reference sites with similar elevations (particularly between S2 and R2, and S3 and R3; Table 1), but other physical parameters were similar at those sites.

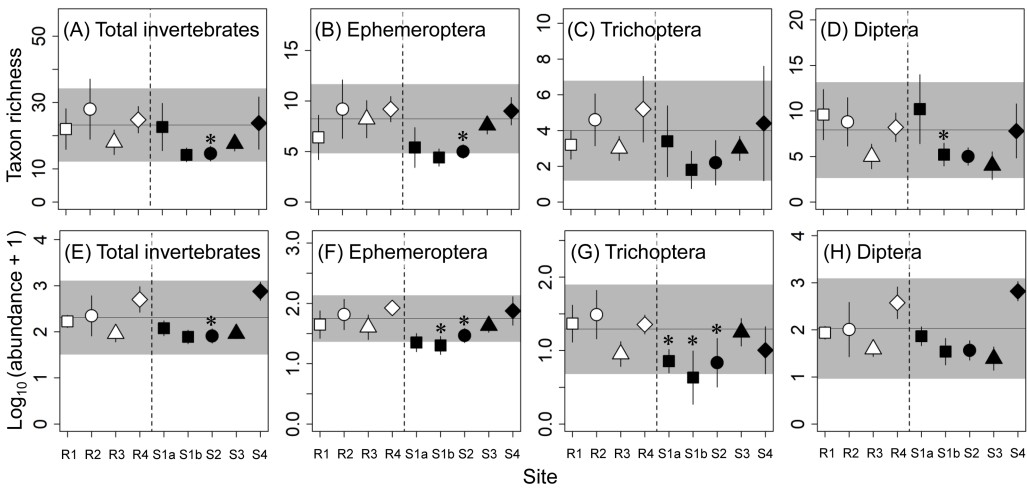

**Figure 2  Taxon richness (number of taxa; A–D) and abundance (number of individuals; E–H) of macroinvertebrates at reference (R1–R4) and contaminated (S1a–S4) sites.** The same symbols indicate sites with similar elevations. Error bars indicate 90% confidence intervals of site means. Horizontal lines and gray areas are the means and 90% prediction intervals calculated from means for the four reference sites, respectively. Asterisks indicate contaminated sites with values significantly lower or higher than the corresponding reference sites with similar elevation ($P < 0.05$).

## Macroinvertebrates and fishes

All eight community metrics for macroinvertebrates at S3 and S4 were within the reference ranges and were not significantly different from those at the corresponding reference sites (Fig. 2), indicating that there were no effects of concern at those contaminated sites. On the other hand, there were adverse effects or some effects of concern for several of the community metrics at the upstream contaminated sites (S1a, S1b, and S2). For example, the mayfly abundance at S1b (58% lower than at R1) and the caddisfly abundance at S1b (83% lower that at R1) were lower than the reference ranges and significantly lower than at the corresponding reference sites.

As with the metrics for the macroinvertebrate community, there were no effects of concern for the abundances of any of the six dominant macroinvertebrate families at S3 and S4 (Fig. 3). Although the variations within individual sites (i.e., the 90% confidence intervals of site means) were relatively large, the abundance of heptageniid mayflies at S1b (68% lower than R1) and the abundance of hydropsychid caddisflies at S1a (84% lower than R1) were lower than the reference ranges and significantly lower than at the corresponding reference sites, indicating adverse effects. Furthermore, there were some effects of concern for the abundances of Simuliidae and Chironomidae at some of the upstream contaminated sites (S1a, S1b, and S2).

No adverse effects were detected for the abundances or condition factors of the four fish species sampled, except for the abundance of masu salmon at S3. Although there were some occasional effects of concern (e.g., the abundances of white-spotted char at S1a and S1b and *B. oreas* at S2 and S3; Fig. 4), the sites where significant differences were observed or the mean value was higher or lower than the reference range varied depending

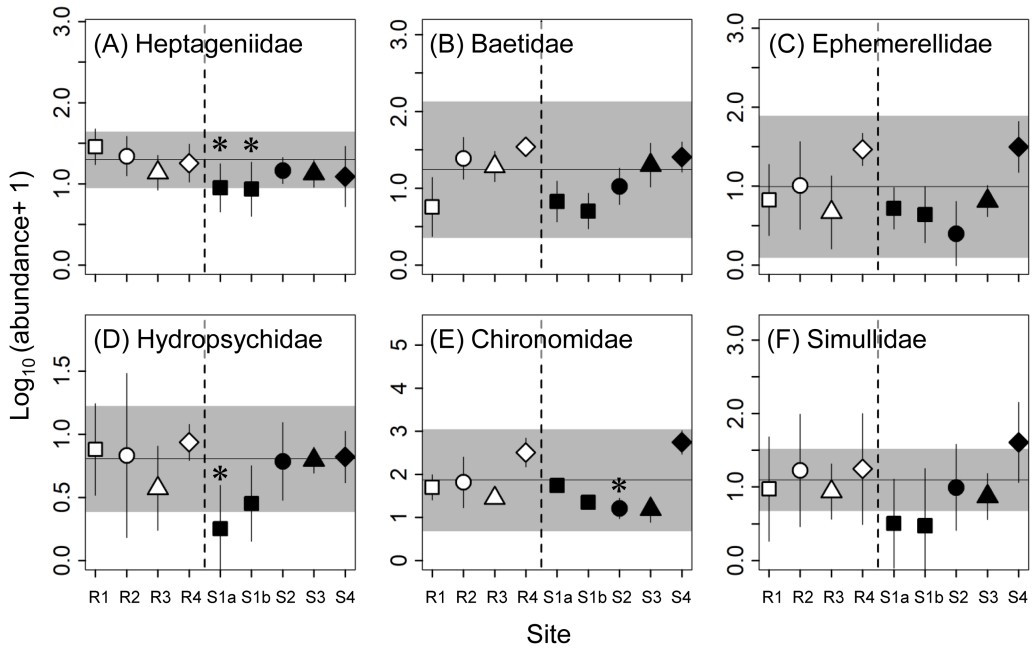

**Figure 3** **Abundance (number of individuals per stone) of dominant families (A–F) of macroinvertebrates at reference (R1–R4) and contaminated (S1a–S4) sites.** The same symbols indicate sites with similar elevations. Error bars indicate 90% confidence intervals of site means. Horizontal lines and gray areas are the means and 90% prediction intervals calculated from means for the four reference sites, respectively. Asterisks indicate contaminated sites with values significantly lower or higher than the corresponding reference sites with similar elevations ($P < 0.05$).

on species. Lamprey (*Lethenteron* spp.) were not collected at most of the contaminated sites (i.e., S1a–S3), but the numbers of lamprey collected were also limited at the reference sites (R2–R4; a total of 5–15 individuals per site) and their variations were relatively large (see Fig. 4). An adverse effect was detected for the abundance of masu salmon at S3, whereas there were no effects of concern for this metric at the other contaminated sites. The estimated proportions of released hatchery masu salmon at three of the reference sites (R1, R3, R4) and two of the contaminated sites (S1a, S1b) were 0%, whereas at R2, S2, S3, and S4 the proportions were 9% (2 of 23), 48% (13 of 27), 5% (1 of 21), and 18% (4 of 22), respectively. We estimated the abundances of wild masu salmon at each site using these proportions and reran the two analyses. The reanalysis did not change the conclusions on the effects of mine contamination on the abundance of masu salmon at contaminated sites.

## DISCUSSION

Our results suggest that macroinvertebrate communities and fish populations at the two downstream sites in the contaminated river in northern Japan, with CCU values <4, were little affected by metal contamination. This is consistent with the results of a previous study in 2017 sampling benthic macroinvertebrates (see *Iwasaki et al. (2020b)* for the detailed discussion about the relationship between CCUs and effects on macroinvertebrate richness

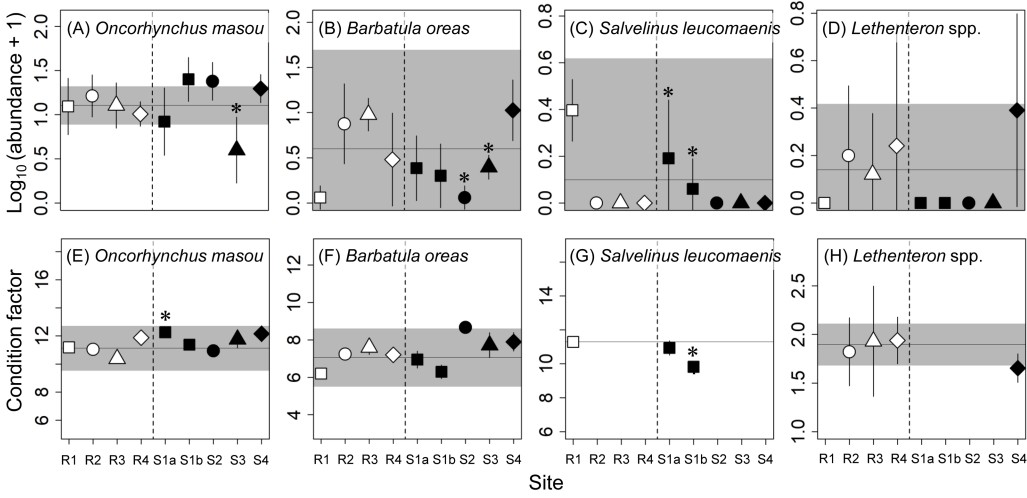

**Figure 4** **Abundance (number of individuals per 50 m$^2$; A–D) and condition factor (E–H) of fishes at reference (R1–R4) and contaminated (S1a–S4) sites.** The same symbols indicate sites with similar elevations. Error bars indicate 90% confidence intervals of site means. Horizontal lines and gray areas are the means and 90% prediction intervals calculated from means for the four reference sites, respectively. Asterisks indicate contaminated sites with values significantly lower or higher than the corresponding reference sites with similar elevations ($P < 0.05$). For *S. leucomaenis*, the 90% prediction interval was not calculated from reference site means because this species was only captured at one reference site (R1).

and abundance). Although we observed a significant decrease in the abundance of masu salmon at S3, this is unlikely due to metal contamination because no such decrease was observed at the contaminated sites farther upstream with higher metal concentrations (Fig. 4).

The concentration of dissolved Zn at the most upstream reference site (R1; Table 2) was relatively high compared with other reference sites and the US EPA WQC (the CCU value was 2.1 at this site). The relative standard deviation for Zn based on three replicate water samples was small (2%) at R1. Although there were no measurements before the sampling campaign, the Zn concentration at R1 was comparable to other reference sites in the sampling conducted in September 2018 (1.0 µg/L; Table S1). It is impossible to determine the underlying reasons for the relatively high Zn concentration at R1, but it is reasonable to regard R1 as a reference site given that we detected no effects on macroinvertebrates and fishes at S3 and S4 with CCUs <4. Furthermore, although our results on metal concentrations and CCUs were based on single-occasion grab samples, the metal concentrations at the study sites were generally similar to those in the field studies performed in June 2017 (*Iwasaki et al., 2020b*) and September 2018 (Table S1). Also, the mean concentrations (±standard errors) of total Zn were 24 (±2) at S2 (five samples in May–October, 2017) and 10 (±2) at S4 (10 samples in May–February, 2017) (*Hokkaido Prefecture, 2018*). As was also observed by *Iwasaki et al. (2020b)*, these results provide modest support that our measurements during the field sampling can be regarded as approximate annual means of metal concentrations in the studied river.

At the two upstream sites (S1a and S1b) with CCU values of approximately 9, we detected adverse effects with some macroinvertebrate metrics, such as the mayfly abundance and the abundance of heptageniid mayflies. Similar results were obtained in the benthic macroinvertebrate sampling in September 2018 (Fig. S1 and S2). Among the macroinvertebrate metrics, mayfly richness and abundance are relatively sensitive to changes in metal contamination levels (*Carlisle & Clements, 1999*; *Clements, Vieira & Church, 2010*) and heptageniid mayflies are also well known as one of the families most sensitive to metal contamination (*Clements et al., 2000*; *Iwasaki, Schmidt & Clements, 2018*). These results suggest that the metal contamination levels at sites S1a and S1b might have been close to the threshold where some adverse effects on sensitive macroinvertebrates would be detected.

We observed several significantly lower values for some macroinvertebrate metrics at S2 compared with the corresponding reference site (R2), but few effects were observed at S2 in a previous study (*Iwasaki et al., 2020b*) or in the field sampling in September 2018 (Figs. S1 and S2). The lower values at S2 could have been attributable to factors other than metal contamination, given that such lower values in the macroinvertebrate metrics were not often observed at the more upstream sites (S1a and S1b). One possible factor is the presence of stenopsychid caddisflies (3.4 individuals per stone at R2; they were absent at S2). The biomass of macroinvertebrates can increase following colonization of the riverbed by net-spinning stream caddisfly larvae, which construct fixed "retreats" that increase riverbed stability and modify the microhabitat structure (*Takao et al., 2006*; *Nunokawa et al., 2008*; *Statzner, 2012*; *Tumolo et al., 2019*). These stenopsychid caddisfly larvae were collected at S2 and R2 (0.2 and 1.0 individuals per stone, respectively) in the field sampling in June 2017 (*Iwasaki et al., 2020b*) and collected at all the study sites including S2 and R2 in the field sampling in September 2018. Thus, we speculate that the differences in macroinvertebrate metrics between S2 and R2 might have been associated with the presence of stenopsychid caddisflies at R2. Also, given that these stenopsychid caddisflies were found at other metal-contaminated sites (*Iwasaki et al., 2012*) and that the net-spinning hydropsychid caddisflies show intermediate sensitivity to metals (*Iwasaki, Schmidt & Clements, 2018*), the absence of stenopsychid caddisflies at S2 in this study could have resulted from factors other than metal contamination. While biological assessments like this study are useful for detecting ecological impairments in the field (*Barbour et al., 1999*), diagnostic tests for metal exposure and biomarkers may be valuable to further examine their causes (*Forbes, Palmqvist & Bach, 2006*; *Miller et al., 2015*).

With the exception of white-spotted char (*S. leucomaenis*), there were no effects of concern for fish abundances or condition factors, even at the two most contaminated sites (S1a and S1b). Although the abundance and condition factor of white-spotted char at S1a and S1b were significantly lower than at the corresponding reference site, they were still within the reference ranges. Given the relatively large variation and the limited number of individuals collected (a total of 13), further study is likely required to reach a more firm conclusion for this species as well as for lamprey (*Lethenteron* spp.). Results from fish sampling in September 2018 were generally similar to our results (Fig. S3), but there are inconsistencies; the contaminated sites showing significant differences from reference

sites varied between the two sampling periods. However, these results at least suggest that there is little need for concern about the effects of metal contamination on the abundance and condition factor of masu salmon, for which there is a national hatchery at the mouth of the Tokushibetsu River. Our findings should provide valuable information to those concerned about impacts of metal contamination on fishes in the studied river (e.g., the local authority managing the closed mine, recreational anglers, and the fishery agency running the hatchery program).

## CONCLUSIONS

Overall, the results from our field study suggest that the richness and abundance of macroinvertebrates (e.g., abundance of heptageniid mayflies) are more sensitive to metal contamination than the abundance and condition factor of fishes in the river studied. Among macroinvertebrate community metrics, mayfly richness and abundance and heptageniid mayfly abundance have been identified as sensitive (*Carlisle & Clements, 1999*; *Clements et al., 2000*; *Iwasaki, Schmidt & Clements, 2018*). Also, these differences in responses to metal contamination have been reported in several studies, and metrics based on fishes are generally less responsive to metal contamination than those based on macroinvertebrates (*Freund & Petty, 2007*; *Clements, Vieira & Church, 2010*; *Namba et al., 2020*), which is consistent with our results. Although it is difficult to determine the underlying reasons for these differences, spatial–temporal characteristics of organisms' responses to metal contamination should have an important role; macroinvertebrates tend to reflect local and more recent conditions than fishes, which are more mobile and relatively longer-lived. Compared with macroinvertebrates, however, the number of fishes captured and the associated metrics were limited in our study. For instance, benthic fishes such as sculpins can be more responsive to metals than salmonids (*Munkittrick & Dixon, 1989b*; *Maret & MacCoy, 2002*), and physiological and biochemical responses of fishes have been employed as early warnings for the population level effects (*Forbes, Palmqvist & Bach, 2006*; *Hanson, 2009*). It would therefore be valuable to accumulate empirical evidence for ecological indicators sensitive to metal contamination within and among biological groups to choose which groups to survey for general environmental impact assessments in contaminated rivers.

## ACKNOWLEDGEMENTS

This paper does not necessarily reflect the policies or views of any government agencies. We are grateful to Susumu Norota, Tatsushi Miyazaki, and Kazutoshi Ueda for their kind help to conduct the field sampling, and Kaori Nakahata for help with the fish measurements in the laboratory.

### Funding
This study was supported in part by the Environment Research and Technology Development Fund (JPMEERF20185R01) of the Environmental Restoration and Conservation Agency of Japan. The funders had no role in study design, data collection and analysis, decision to publish, or preparation of the manuscript.

### Grant Disclosures
The following grant information was disclosed by the authors:
Environment Research and Technology Development Fund: JPMEERF20185R01.

### Competing Interests
The authors declare there are no competing interests.

### Author Contributions
- Hiroki Namba and Kentaro Morita conceived and designed the experiments, performed the experiments, analyzed the data, prepared figures and/or tables, authored or reviewed drafts of the paper, and approved the final draft.
- Yuichi Iwasaki conceived and designed the experiments, performed the experiments, analyzed the data, prepared figures and/or tables, authored or reviewed drafts of the paper, obtained funding for the study, and approved the final draft.
- Tagiru Ogino and Naohide Shinohara performed the experiments, authored or reviewed drafts of the paper, and approved the final draft.
- Hiroyuki Mano conceived and designed the experiments, performed the experiments, authored or reviewed drafts of the paper, obtained funding for the study, and approved the final draft.
- Tetsuo Yasutaka conceived and designed the experiments, authored or reviewed drafts of the paper, obtained funding for the study, and approved the final draft.
- Hiroyuki Matsuda conceived and designed the experiments, authored or reviewed drafts of the paper, and approved the final draft.
- Masashi Kamo performed the experiments, authored or reviewed drafts of the paper, obtained funding for the study, and approved the final draft.

### Animal Ethics
The following information was supplied relating to ethical approvals (i.e., approving body and any reference numbers):

No relevant regulations for fishes collected from the field are available in Japan. Also, except for some fish for checking the otolith marks, all other fishes were released.

### Field Study Permissions
The following information was supplied relating to field study approvals (i.e., approving body and any reference numbers):

Field sampling in the studied river was approved by the local municipal office (Esashi Town, Hokkaido). Fish collection was approved by the local Hokkaido government. Except for some fish collected to check the otolith marks, all other fishes were released.

## Data Availability

All the data used are available in the Supplemental Files.

## Supplemental Information

Supplemental information for this article can be found online at http://dx.doi.org/10.7717/peerj.10808#supplemental-information.

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
