# Peer review of "Comparing impacts of metal contamination on macroinvertebrate and fish assemblages in a northern Japanese river"

_PeerJ, doi:10.7717/peerj.10808_

## Round 0.1 · original submission · Major Revisions

The reviewers all agree that your paper has merit, but have indicated significant areas which need attention. Reviewer 2 is most concerned about your decision to exclude study data (Sept 2018) from your analysis and I agree that this should be addressed in your revised manuscript. Also, some additional coverage of the implications of your results for fisheries would be valuable.

·

Basic reporting

The manuscript was written in a clear, concise style throughout. The introduction presented well the state of the art regarding metal impacts on freshwater ecosystems and then focussed down on the study objective; which biological groups are best suited to assessment of metal impacts. My only minor criticism is that the authors give the impression that the study will be focussing on the impact of the treated discharge from an abandoned mine when in fact there are other upstream sources of metal contamination upstream; a point already confirmed in a previous paper by the authors.
The manuscript is well-structured throughout making it easy to understand and follow the steps taken in the research. While a precise experimental hypothesis is not stated the authors do clearly define the objective of the survey.

Experimental design

The authors describe a simple comparison between two adjacent streams, one in a catchment with an abandoned mine facility and the other in a catchment without. Multiple discrete sites are sampled on a single occasion down the length of both streams. There is within-site replication of water samples, macroinvertebrate samples and fish sampling, and replication of sites within each catchment, but there is neither temporal replication at each site nor replication at the level of catchment i.e. sampling of multiple mining-impacted and non-impacted catchments. The lack of replication at the two latter levels constrains the work to being a case-study of impacts in the Ofuntarumanai River relative to the Tokushibetsu River. The lack of temporal replication of water chemistry sampling is particularly concerning given that it is well-understood how variable such measurements can be in relation to discharge. A single measurement can only provide a crude indication of the water quality conditions experienced by the biota over the preceding weeks and months.
It would be good to have more detail on the geology and land-use of the two catchments to confirm that the reference stream is an appropriate comparator for the impacted stream.
The biological sampling and processing, chemical sampling and analysis are well-described with sufficient detail. The data analysis steps are also clearly detailed, though I do raise a query regarding the use of multiple comparison tests prior to ANOVA. In my experience it has always been the other way round; you do the ANOVA first and then carry out multiple comparison texts to determine where the significant differences lie. It would be useful if the authors could provide more information on this point.

Validity of the findings

The authors are careful to limit their statement of results to the constraints of the study design. They find that the abundance and condition of four fish species did not differ consistently between the two catchments, while some descriptors of macroinvertebrate communities at the more upstream sites did differ from those in the reference catchment and upstream reference sites. Given the lack of catchment replication they can only suggest that metal concentrations are the limiting factor at play in the upstream sites.
Another minor criticism is that the current manuscript is effectively a repeat of the Iwasaki et al 2020 work based on a similar sampling survey undertaken in 2017 only this time they have included a fish survey at each site. The authors claim that the comparison of fish and macroinvertebrate responses to metal contamination will contribute to the ‘surprisingly…limited number of studies published in peer-reviewed journals [that] have investigated multiple biological groups in metal-contaminated rivers’. But in the conclusions they state that ‘these differences in responses to metal contamination have been reported in several studies, and metrics based on fishes are generally less responsive to metal contamination than those based on macroinvertebrates’. So this manuscript would appear to be effectively an annual update of the findings from Iwasaki et al 2020 with the addition of fish data to confirm what is already known i.e. fish are less responsive as bioindicators of metal pollution than macroinvertebrates. Furthermore the authors include data for an autumn 2018 survey of the same streams in the discussion but not formally as part of the paper. One can’t help thinking that perhaps the 2017-2018 datasets would be better analysed as one body of data in a single manuscript.

Additional comments

As stated above I found the manuscript to be very well written and presented. My main concern is that it is based on a limited dataset with little to no effective replication. This hampers your ability to draw general conclusions of wider relevance beyond the Tokushibetsu catchment.
I have uploaded an annotated copy of the manuscript pdf with some additional comments.

Reviewer 2 ·

Basic reporting

See attachment.

Experimental design

See attachment.

Validity of the findings

See attachment.

Additional comments

This is an important field study that provides an important contribution to our understanding of stream community responses to mining impacts and remediation in streams. Most of my comments request additional information and clarification. However, my most outstanding comment/concern is that field data from Sept-2018 were not part of the manuscript's primary analyses. The central hypothesis of the study was to test for the ecological effects of stream communities to treated mine discharge in the Tokushibetsu River. Therefore, it makes no sense to exclude the Sept-2018 data from the primary statistical analysis and to just briefly refer to these data in the discussion. I strongly suggest modeling both June and September data together. I provide more detail regarding this concern in the attached file.

Annotated reviews are not available for download in order to protect the identity of reviewers who chose to remain anonymous.

Reviewer 3 ·

Basic reporting

This manuscript explores the effects of metal contamination on aquatic macroinvertebrates and fish in a stream system receiving treated discharge from a decommissioned mine in northern Japan. The paper is in relatively good shape. One of the proposed unique features of this study is that multiple biological groups (BMIs and fish) are studied together to evaluate whether there are any differences in the sensitivity of metrics calculated for macroinvertebrates and for fish communities.

I commend the authors for writing a clear and concise manuscript to read. However, a number of edits are required and I would like to see certain sections expanded. Please find specific edit requests below. The Raw data is provided and is easy to interpret. The figures are simple and well labelled. The figure captions are clear. A permit has been supplied in the supplementary information.

Specific edits:

Line 30-31. Should this read “At the two most (or furthest) upstream contaminated sites...”? It is unclear what sites you are referring to from the study design because S1a is technically upstream of the discharge area, so this could be either S1A and S1B or S1B and S2? Please clarify.

Line 30 to 33. Restructure this sentence to begin with the result (“We detected a significant (observation) at the two upstream contaminated sites with metal concentrations....”.

Line 35. Include scientific name in brackets within the abstract unless journal guidelines suggest otherwise.

Line 90. Relabel Table 2 as Table 1 and reference it here to guide readers to your m.a.s.l values.

Line 253, 259, 262 and throughout. You switch between O. masou and masu salmon throughout. Please be consistent.

Experimental design

This paper is within the scope of the journal and makes an interesting contribution to the field. In general, the objective is defined concisely, but I would like to see further development regarding the knowledge gap and uniqueness of the study (see specific comments for Line 69 and Line 73 below). The statistical design is simple and effective; however, I would consider further multivariate analyses relating macroinvertebrate communities to water chemistry metrics to help accumulate empirical evidence for the macroinvertebrate indicators

Specific edits:
Line 69-70. I think one additional component that makes your study unique is related to your investigation of legacy effects/long-term effects post-closure of the mine. Please strengthen this component in your introduction. In addition, is the river passively receiving discharge from the artificial wetlands or are periodic discharge events occurring throughout the year to the river system?

Line 73 to 75. I would like to see further justification for the inclusion of fish in this assessment from an ecosystem service perspective given that it is one of the defining features that differentiates this study from the 2017 assessment (Iwasaki et al. 2020) referenced in line 72. Is this a catch and release fishery, is it important food resource for local communities, etc.? Also, is this the only important fish species in the region? Why not just focus on masu salmon instead of other species? Are there any additional taxa that should have been included that are sensitive to metal contamination that are present in this system that were overlooked (e.g. sculpin)? Please develop your reasoning in this paragraph.

Line 84 to 92. Please include 1-2 sentences describing the general characteristics of the catchment. Are other disturbance types an issue here? Is this catchment developed or more rural? These details would be helpful to an international audience

Line 86 to 88. Can you include stream order?

Line 86. Is S1A a true contaminated site while technically being upstream of the output area? I noticed in a previous investigation (Iwasaki et al. 2020 Environmental Monitoring and Assessment), this site was used in an upstream/downstream comparison of the effects of metal contaminated discharge. I think including justification for S1A as a contaminated site would be prudent.

Line 86 to 91. Are the river systems geographically distant enough that you would not expect migration of fish between the contaminated and reference study reach?

Line 139 to 146. Please add a sentence describing average substrate type in your reaches. Are these sites typically rock/cobble-dominated, therefore justifying your method of surber sampler and rock scraping?

Line 170 to 175. Are Masu salmon stocked in both river reaches? Do you know when they are stocked and how many are stocked? This could artificially inflate abundance estimates at one reach, and depending on how long they have been in the river system, may influence condition.

Line 196. The statistics are sufficiently communicated and I do enjoy the simplicity of the analyses, however, were multivariate analyses considered here? Given the data that you have including water chemistry and physical variables, I see potential to investigate whether there are environmental drivers of community structure at your sites using redundancy analysis or similar appropriate ordination technique. This is just a suggestion, but it may provide a deeper understanding than possible with your current metric comparison. In addition, a similarity percentage analysis (SIMPER) may help investigate potential differences between the R and S sites.

Table 1. Relabel as Table 2.

Table 2. Relabel as Table 1 now that it is referenced in Line 90.

Validity of the findings

All underlying data has been provided, analysis are simple and effective, and results are statistically sound. Speculation is kept to a minimum. I would recommend adding a few sentences to emphasize the long-term detectable patterns observed in macroinvertebrate community metrics related to the mining operation. Considering the localized context of this paper, I would like to see some discussion of the importance of these findings to the region and stocked population of masu salmon. Masu salmon stocking appears to be one of the points raised to argue the importance of this study in the introduction (Lines 73-74). I also noticed that a section of the discussion (Lines 308 to 313) mention significant differences between two of the most upstream impacted sites compared to upstream references sites for white spotted char (S. leucomaenis) and a trend is also seen that appears important enough to mention, although it’s not significant, for Lethenteron spp. I would recommend including a sentence in the results section to prepare the reader for this discussion, as it is currently lacking. Relevant references are used throughout the discussion that support the findings in this study.

---

## Round 0.2 · accepted · Accept

Both reviewers are satisfied with your revisions, and I am happy to accept the manuscript as revised.

·

Basic reporting

The manuscript now meets the journal standards for basic reporting

Experimental design

The manuscript now meets the journal standards with regard to its survey design and data analysis.

Validity of the findings

The manuscript now meets the journal standards with regard to the validity of the findings

Reviewer 2 ·

Basic reporting

No Comment

Experimental design

The authors have addressed my overall biggest concern regarding the exclusion of September field sampling data from their primary analyses. Based on the previous reporting of the September sampling I was unaware that different benthic sampling methods were used and without replication at many sites. This is an important detail and now I agree with the decision to not include these data in the primary analyses. However, given these data are publicly available and are summarized in supporting information, I believe it is fine to reference these data in the discussion section of the manuscript.

Validity of the findings

The previous comment in the experimental design section links to the validity of the findings. In short, I believe the authors have addressed all of my concerns, the results are valid and the discussion and conclusions are well supported.

Additional comments

The authors did a very good job addressing my suggested revisions. My recommendation is that this manuscript is accepted for publication.